# SOAP: Enhancing Spatio-Temporal Relation and Motion Information Capturing for Few-Shot Action Recognition

Wenbo Huang
Southeast University
Nanjing, China
wenbohuang1002@outlook.com

Jinghui Zhang*
Southeast University
Nanjing, China
jhzhang@seu.edu.cn

Xuwei Qian
Southeast University
Nanjing, China
xuwei.qian@seu.edu.cn

Zhen Wu
Southeast University
Nanjing, China
zhen-wu@seu.edu.cn

Meng Wang
Tongji University
Shanghai, China
wangmengsd@outlook.com

Lei Zhang
Nanjing Normal University
Nanjing, China
leizhang@njnu.edu.cn

## Abstract

High frame-rate (HFR) videos of action recognition improve fine-grained expression while reducing the spatio-temporal relation and motion information density. Thus, large amounts of video samples are continuously required for traditional data-driven training. However, samples are not always sufficient in real-world scenarios, promoting few-shot action recognition (FSAR) research. We observe that most recent FSAR works build spatio-temporal relation of video samples via temporal alignment after spatial feature extraction, cutting apart spatial and temporal features within samples. They also capture motion information via narrow perspectives between adjacent frames without considering density, leading to insufficient motion information capturing. Therefore, we propose a novel plug-and-play architecture for FSAR called **S**patio-temp**O**ral fr**A**me tu**P**le enhancer (**SOAP**) in this paper. The model we designed with such architecture refers to SOAP-Net. Temporal connections between different feature channels and spatio-temporal relation of features are considered instead of simple feature extraction. Comprehensive motion information is also captured, using frame tuples with multiple frames containing more motion information than adjacent frames. Combining frame tuples of diverse frame counts further provides a broader perspective. SOAP-Net achieves new state-of-the-art performance across well-known benchmarks such as SthSthV2, Kinetics, UCF101, and HMDB51. Extensive empirical evaluations underscore the competitiveness, pluggability, generalization, and robustness of SOAP. The code is released at https://github.com/wenbohuang1002/SOAP.

## CCS Concepts

• **Computing methodologies → Activity recognition and understanding**.

*Corresponding authors: Jinghui Zhang

## Keywords

Action recognition; few-shot learning; spatio-temporal relation; motion information

**ACM Reference Format:**
Wenbo Huang, Jinghui Zhang, Xuwei Qian, Zhen Wu, Meng Wang, and Lei Zhang. 2024. SOAP: Enhancing Spatio-Temporal Relation and Motion Information Capturing for Few-Shot Action Recognition. In *Proceedings of the 32nd ACM International Conference on Multimedia (MM '24), October 28–November 1, 2024, Melbourne, VIC, Australia.* ACM, New York, NY, USA, 9 pages. https://doi.org/10.1145/3664647.3681062

## 1 Introduction

Ubiquitous videos in daily lives are rapidly accelerating the development of multimedia analytic research. As a fundamental task, action recognition is experiencing an explosive demand in a wide range of applications including intelligent surveillance, video understanding, and health monitoring [4, 15, 20]. Progress in video recorders contributes to high frame-rate (HFR) videos, with more similar frames per second improving the expression of fine-grained actions [17]. As the example shown in Figure 1, we can explicitly observe that the timeline and displacement of object in HFR video frames are much subtler than those in low frame-rate (LFR) video frames, better reflecting fine-grained actions. However, the spatio-temporal relation and motion information density decrease with the improvement of video fluency [36]. Therefore, a larger amount of video samples are continuously required to train data-driven models. Unfortunately, samples for target actions such as "falling down" are usually insufficient and hard to collect in real-world scenarios. Contemporary few-shot learning mitigates data dependence by transferring knowledge from a few samples, promoting few-shot action recognition (FSAR) research.

According to data characteristics of HFR videos, two prevailing challenges of FSAR exist. *Challenge 1: Optimizing spatio-temporal relation construction.* Spatial and temporal features work as a whole in video samples, only focusing on spatial information makes models misidentify horizontal or vertical actions such as "pushing", "pulling", "putting up", and "putting down". However, the spatio-temporal relation of HFR videos is subtle, making the construction challenging. *Challenge 2: Comprehensive motion information capturing.* As an exclusive characteristic of videos, motion information plays a crucial role in helping models recognize target actions in a dynamic manner. However, the difficulties in motion information

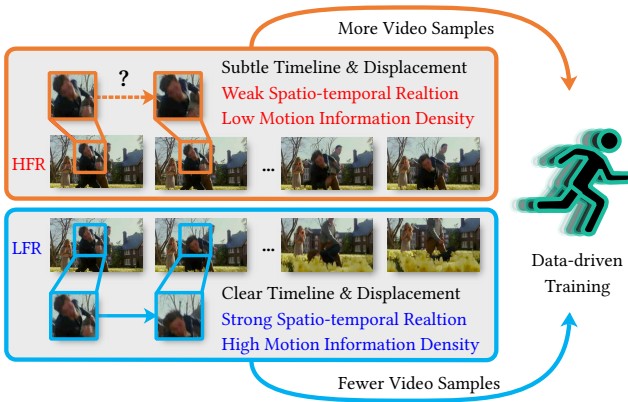

**Figure 1: Spatio-temporal relation and motion information density of HFR video frames are much subtler, reflecting by timeline and displacement. Therefore, larger amounts of samples are required for data-driven training.**

capturing is exaggerated by low motion information density from HFR videos and limited frames processed at one time with mainstream methods. In summary, the challenges mentioned above not only stem from HFR videos but are also aggravated by the insufficient samples in few-shot settings.

In FSAR, the metric-based paradigm predominates due to its simplicity and efficacy. It embeds samples into action prototypes to calculate support-query distances for classification in an episodic task [1, 18, 30, 41, 43]. We observe that these works prioritize temporal alignment after spatial feature extraction, cutting apart spatial and temporal features within samples and overlooking the importance of motion information capture. Some works [29, 31, 33] incorporate motion information into action recognition and achieve remarkable performance. However, the motion information is captured between adjacent frames. This narrow perspective inevitably overlooks density of motion information and results in inadequate capturing. So far, no solution exists for both challenges.

Motivated by aforementioned challenges, we propose a novel plug-and-play architecture for FSAR called **S**patio-temp**O**ral fr**A**me tu**P**le enhancer (**SOAP**). The model we designed with such architecture is defined as SOAP-Net. The cores are optimizing construction of spatio-temporal relation and capturing comprehensive motion information. For the first goal, simply extracting features from video frames is insufficient. These features are located in different channels and have temporal connections between each channel. Spatio-temporal relation within features also play a key role in FSAR. For the second goal, we consider motion information density and find that frame tuples with multiple frames contain richer motion information than adjacent frames. Combining frame tuples of various frame counts further provides a broader perspective. To be specific, SOAP has three components: 3-Dimension Enhancement Module (3DEM) uses a 3D convolution for spatio-temporal relation construction and Channel-Wise Enhancement Module (CWEM) calibrates channel-wise feature responses adaptively while Hybrid Motion Enhancement Module (HMEM) applies a broader perspective to help models capture comprehensive motion information.

These three modules work in parallel, adding priors into raw input.

To our best knowledge, SOAP is the first to address all challenges simultaneously. Our main contribution are three-fold:

- **Spatio-Temporal Relation Construction.** The proposed SOAP optimizes spatio-temporal relation construction, avoiding simply operating temporal alignment after extracting spatial features.
- **Motion Information Capturing.** Taking motion information density and the processing method into account, our SOAP finds a comprehensive solution, overcoming the second challenge with a broader perspective that combines frame tuples with various frame counts.
- **Effectiveness.** SOAP-Net achieves new SOTA performance on several well-known FSAR benchmarks, including Sth-SthV2, Kinetics, UCF101, and HMDB51. Comprehensive experiments demonstrate the competitiveness, pluggability, generalization, and robustness of SOAP.

## 2 Related Works

### 2.1 Few-Shot Learning

The core goal of few-shot learning is to recognize unseen classes from only few samples. Unlike traditional deep learning, few-shot learning utilizes episodic training where training units are structured as similar tasks with small labeled sets. Three primary groups of few-shot learning are data-augmentation based, optimization-based, and metric-based paradigms. In the first data-augmentation paradigm, generating additional samples to supplement available data is a key feature. Specifically, MetaGAN [39] utilizes generative adversarial networks (GANs) and statistics of existing samples to synthesize data. The most representative optimization-based method is MAML [7], which identifies a model parameters set and adapts it to individual tasks via gradient descent. The metric-based paradigm is well-known and widely used for its simplicity and effectiveness. It leverages similarity between support samples to classify query samples into corresponding classes. Prototypical Networks [22] construct prototypes based on class centroids and then classify samples by measuring distance to each prototype. Most few-shot learning paradigms are applied to image classification, with fewer in the field of videos. Our SOAP belongs to the metric-based paradigm for FSAR, focusing on improving prototype representation ability.

### 2.2 Action Recognition

Compared to image classification, action recognition is a more complex and extensively researched problem in the community due to the spatio-temporal relation and motion information. Previous works [3, 19, 26, 32] have typically utilized 3D backbones to construct spatio-temporal relations, while optical flow is applied by additional networks to inject video motion information for action recognition, resulting in promising results. However, these works with high acquisition costs are all designed for traditional data-driven training without considering insufficient samples in real-world scenarios. In contrast, SOAP is specifically designed for a more realistic few-shot setting.

## 2.3 Few-Shot Action Recognition

Existing methods of FSAR mainly focus on metric-based paradigm for effective prototype construction. Among them, CMN [43, 44] introduces a multi-saliency embedding algorithm for key frame encoding, improving the prototype representation; ARN [38] captures short-range dependencies using 3D backbones with a self-trained permutation invariant attention; OTAM [1] aligns support and query with a dynamic time warping (DTW) algorithm. Joint spatio-temporal modeling approaches such as TA$^2$N [13], ITA [40], STRM [24], and SloshNet [34] employ spatial-temporal frameworks to address the support-query misalignment from multiple perspectives; Temporal relation is emphasized by TRX [18], using a Cross Transformer for sub-sequence alignment and achieving notable improvement; HyRSM [30] learns the task-specific embedding by exploiting the relation within and cross videos; SA-CT [41] complement the temporal information by learning the spatial relation. The temporal alignment all operated after spatial feature extraction. Diverse types of information including optical flow [31] and depth [8] are introduced by extra networks, increasing computations. MoLo [29] and MTFAN [33] explicitly extract motion information from raw videos and leverage temporal context for FSAR. Although achieving promising performance, these works are either cut apart spatial and temporal features or ignore motion information density. Therefore, our SOAP focuses on spatio-temporal relation and motion information capturing.

## 3 Methodology

### 3.1 Problem Formulation

Classifying an unlabeled query into one of the support classes is the inference goal of FSAR. The support set has at least one labeled sample per class. Following the prior works [1, 18, 29, 30], we randomly select few-shot tasks from training set for episodic training. In each task, the support set $S$ contains $N$ classes and $K$ samples per class, *i.e.*, $N$-way $K$-shot setting. The query set $Q$ in each task has samples to be classified. Uniformly sampling $F$ frames of each video, we define the $k^{\text{th}}$ ($k = 1, \cdots, K$) sample of the $c^{\text{th}}$ ($c = 1, \cdots, N$) class of support set $S$ as:

$$S^{ck} = \left[ s_1^{ck}, \ldots, s_F^{ck} \right] \in \mathbb{R}^{F \times C \times H \times W}. \tag{1}$$

Notions used are $F$ (frames), $C$ (channels), $H$ (height), and $W$ (width), while the sample with $F$ frames of query is defined as:

$$Q = [q_1, \ldots, q_F] \in \mathbb{R}^{F \times C \times H \times W}. \tag{2}$$

Models of FSAR need to predict labels for query samples with the guidance of the support set.

### 3.2 Overall Architecture

An overview of SOAP-Net is provided in Figure 2. For clear description, we take the $Q$ and $c^{\text{th}}$ of $S$ as an example to illustrate the whole process of our method. First, video samples are decoded into frames at fixed intervals to serve as the input. Then, support and query are sent into three main modules of SOAP, *i.e.*, 3DEM, CWEM, and HMEM. To be specific, 3DEM establishes spatio-temporal relation of features, while CWEM adaptively calibrates temporal connections between channels. Instead of solely concentrating on motion

information between adjacent frames, HMEM adopts a broader perspective on frame tuples of varying frame counts, delivering comprehensive motion information through a hybrid approach. The three modules are arranged in parallel to generate triple prior guidance before feature extraction. Several linear layers are subsequently adopted for prototype construction. Finally, the distance between query and prototype can be calculated for classification.

### 3.3 3-Dimension Enhancement Module

For the relation construction between spatial and temporal information, 3DEM is designed based on 3D convolution. The structure of this module is shown in Figure 3. Firstly, 3DEM averages the support and query across the channels, resulting in spatio-temporal tensors. Then the corresponding tensors are reshaped. The above operation can be formulated as:

$$\tilde{S}_1^{ck} = \text{Reshape}_1 \left( \frac{1}{C} \sum_{g=1}^{C} S^{ck}(:, g, :, :) \right) \in \mathbb{R}^{1 \times F \times H \times W},$$
$$\tilde{Q}_1 = \text{Reshape}_1 \left( \frac{1}{C} \sum_{g=1}^{C} Q(:, g, :, :) \right) \in \mathbb{R}^{1 \times F \times H \times W}. \tag{3}$$

A 3D convolution is used for constructing the spatio-temporal relation and another reshape operation is used for shape recovery:

$$\hat{S}_1^{ck} = \text{Reshape}_2 \left( \text{Conv3D} \left( \tilde{S}_1^{ck} \right) \right) \in \mathbb{R}^{F \times 1 \times H \times W},$$
$$\hat{Q}_1 = \text{Reshape}_2 \left( \text{Conv3D} \left( \tilde{Q}_1 \right) \right) \in \mathbb{R}^{F \times 1 \times H \times W}. \tag{4}$$

Finally, spatio-temporal relation are fed into a Sigmoid activation and residually connected with the input to generate the 3D prior knowledge before feature extraction:

$$S_1^{ck} = S^{ck} + \text{Sigmoid} \left( \hat{S}_1^{ck} \right) \times S^{ck},$$
$$Q_1 = Q + \text{Sigmoid} \left( \hat{Q}_1 \right) \times Q. \tag{5}$$

For each input channel, 3DEM can assist the spatio-temporal relation construction.

### 3.4 Channel-Wise Enhancement Module

Features located in different channels have temporal connections between each channel. Inspired by SE [11], we design this module with the addition of a simple 1D convolution. The structure of CWEM is detailed in Figure 4. First, a spatial average pooling is performed on the support and query, followed by a 2D convolution that expands the channel number to $C_r$, represented as:

$$\tilde{S}_2^{ck} = \text{Conv2D}_1 \left( \frac{1}{H \times W} \sum_{i=1}^{H} \sum_{j=1}^{W} S^{ck}(:, :, i, j) \right) \in \mathbb{R}^{F \times C_r \times 1 \times 1},$$
$$\tilde{Q}_2 = \text{Conv2D}_1 \left( \frac{1}{H \times W} \sum_{i=1}^{H} \sum_{j=1}^{W} Q(:, :, i, j) \right) \in \mathbb{R}^{F \times C_r \times 1 \times 1}. \tag{6}$$

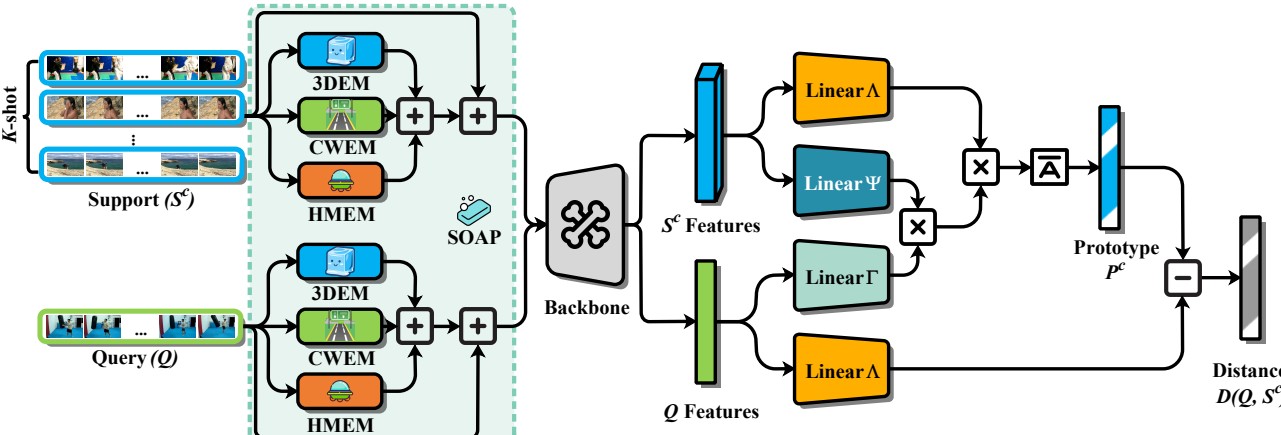

**Figure 2: Overview of the SOAP-Net. It comprises three main modules: the 3DEM for constructing relation between spatial and temporal information, the CWEM for modeling temporal connections between channels, and the HMEM for capturing comprehensive motion information with frame tuples of varying frame counts using a hybrid approach. The "Ā" symbol at the right part of the figure shows an averaging calculation used to construct a query-specific prototype $P^c$ in Eqn (17).**

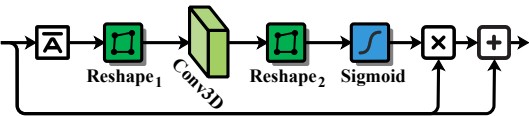

**Figure 3: The structure of 3-Dimension Enhancement Module (3DEM). The "Ā" at left part means the averaging calculation in Eqn (3).**

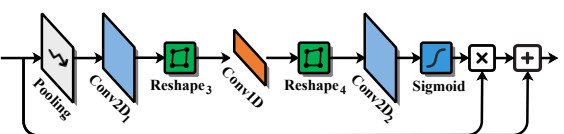

**Figure 4: The structure of Channel-Wise Enhancement Module (CWEM).**

The third reshape operation prepares for the 1D convolution, which adaptively calibrates channel-wise feature responses, as:

$$\hat{S}_2^{ck} = \text{Conv1D}\left(\text{Reshape}_3\left(\tilde{S}_2^{ck}\right)\right) \in \mathbb{R}^{C_r \times F \times 1},$$
$$\hat{Q}_2 = \text{Conv1D}\left(\text{Reshape}_3\left(\tilde{Q}_2\right)\right) \in \mathbb{R}^{C_r \times F \times 1}. \tag{7}$$

Once channel-wise feature responses are calibrated, the next steps are reshape and 2D convolution. Together, these operations recover the original dimension and channel numbers of the feature maps. This process can be formulated as follows:

$$\dot{S}_2^{ck} = \text{Conv2D}_2\left(\text{Reshape}_4\left(\hat{S}_2^{ck}\right)\right) \in \mathbb{R}^{F \times C \times 1 \times 1},$$
$$\dot{Q}_2 = \text{Conv2D}_2\left(\text{Reshape}_4\left(\hat{Q}_2\right)\right) \in \mathbb{R}^{F \times C \times 1 \times 1}. \tag{8}$$

The final step of CWEM is formulated similarly to Eqn (5), using Sigmoid and residual connection, to generate channel prior knowledge

before feature extraction:

$$S_2^{ck} = S^{ck} + \text{Sigmoid}\left(\dot{S}_2^{ck}\right) \times S^{ck},$$
$$Q_2 = Q + \text{Sigmoid}\left(\dot{Q}_2\right) \times Q. \tag{9}$$

### 3.5 Hybrid Motion Enhancement Module

In previous works [29, 31], motion information plays a significant role. However, we find that focusing only on the motion information between adjacent frames is insufficient, as subtle displacements are hard to detect. In order to capture more motion information using HMEM, we extend perspective from adjacent frames to frame tuples and apply the combination of multiple scales. The structure of HMEM is demonstrated in Figure 5. We define a set $O$ as a hyperparameter, where the value of element $T$ ($T \in O$, $T < F$ and $T \in \mathbb{N}^*$) represents frame counts of a tuple and the cardinality of set $|O|$ denotes the number of branches. Being achieved by sliding window algorithm $\text{SW}(\cdot, \cdot)$, frame tuple sets of support and query with element index $t$ ($t \in [1, F - T + 1]$) can be represented as:

$$\text{SW}\left(S^{ck}, T\right) = \left[\cdots, \omega_t^S, \omega_{t+1}^S, \cdots\right],$$
$$\text{SW}\left(Q, T\right) = \left[\cdots, \omega_t^Q, \omega_{t+1}^Q, \cdots\right]. \tag{10}$$

We concatenate tuple differences in the first dimension, preparing for motion information calculation. For a simple description, we choose the $e^{\text{th}}$ ($e \in [1, |O|]$ and $e \in \mathbb{N}^*$) branch as an example:

$$M_e^S = \text{Concat}\left(\cdots, \text{Conv2D}^M\left(\omega_{t+1}^S\right) - \omega_t^S, \cdots\right),$$
$$M_e^Q = \text{Concat}\left(\cdots, \text{Conv2D}^M\left(\omega_{t+1}^Q\right) - \omega_t^Q, \cdots\right). \tag{11}$$

Motion information with different scales can be captured by setting multiple branches. Then we concatenate them along the first dimension, achieving tensors belong to $\mathbb{R}^{X \times C \times H \times W}$. Function $Z(\cdot)$

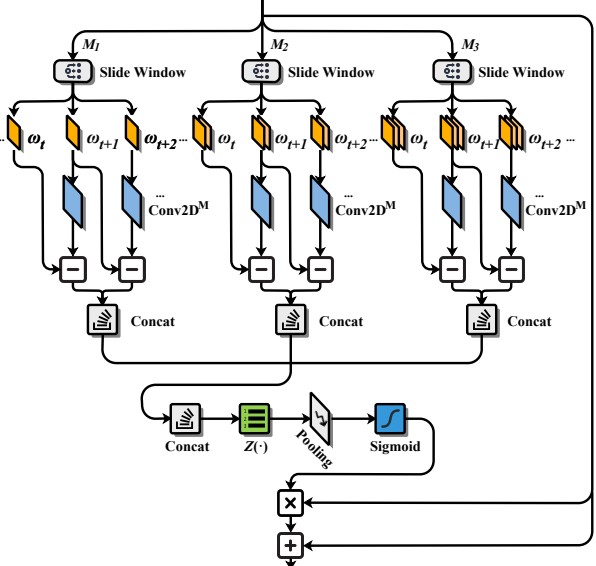

**Figure 5: The structure of Hybrid Motion Enhancement Module (HMEM), where $O = \{1, 2, 3\}$ and "Concat" denotes for concatenate operation.**

that performs flatten ($\mathbb{R}^{X \times C \times H \times W} \mapsto \mathbb{R}^{X \times (C \times H \times W)}$), linear transformation ($\mathbb{R}^{X \times (C \times H \times W)} \mapsto \mathbb{R}^{F \times (C \times H \times W)}$), and reshape operation ($\mathbb{R}^{F \times (C \times H \times W)} \mapsto \mathbb{R}^{F \times C \times H \times W}$) in sequence, acquiring comprehensive motion information. This process is concluded as:

$$\tilde{S}_3^{ck} = Z\left(\text{Concat}\left(\cdots, M_e^S, \cdots\right)\right) \in \mathbb{R}^{F \times C \times H \times W},$$
$$\tilde{Q}_3 = Z\left(\text{Concat}\left(\cdots, M_e^Q, \cdots\right)\right) \in \mathbb{R}^{F \times C \times H \times W}. \tag{12}$$

A spatial average pooling is operated for reducing computation, as:

$$\hat{S}_3^{ck} = \left(\frac{1}{H \times W} \sum_{i=1}^{H} \sum_{j=1}^{W} \tilde{S}_3^{ck}(:,:,i,j)\right) \in \mathbb{R}^{F \times C \times 1 \times 1},$$
$$\hat{Q}_3 = \left(\frac{1}{H \times W} \sum_{i=1}^{H} \sum_{j=1}^{W} \tilde{Q}_3(:,:,i,j)\right) \in \mathbb{R}^{F \times C \times 1 \times 1}. \tag{13}$$

Similar with the prior knowledge calculated by two previous modules, the hybrid motion prior can be represented as:

$$S_3^{ck} = S^{ck} + \text{Sigmoid}\left(\hat{S}_3^{ck}\right) \times S^{ck},$$
$$Q_3 = Q + \text{Sigmoid}\left(\hat{Q}_3\right) \times Q. \tag{14}$$

## 3.6 Prototype Construction

Triple prior guidance are added into the raw support and query before feature extraction:

$$\tilde{S}^{ck} = \left(S_1^{ck} + S_2^{ck} + S_3^{ck}\right) + S^{ck} = \left[\tilde{s}_1^{ck}, \ldots, \tilde{s}_F^{ck}\right] \in \mathbb{R}^{F \times C \times H \times W},$$
$$\tilde{Q} = (Q_1 + Q_2 + Q_3) + Q = [\tilde{q}_1, \ldots, \tilde{q}_F] \in \mathbb{R}^{F \times C \times H \times W}. \tag{15}$$

$\tilde{S}^{ck}$ and $\tilde{Q}$ are sent to the backbone network. Features of support and query are defined as:

$$S_f^{ck} = \left[f_\theta\left(\tilde{s}_1^{ck}\right) + f_{pe}(1), \ldots, f_\theta\left(\tilde{s}_F^{ck}\right) + f_{pe}(F)\right] \in \mathbb{R}^{F \times D},$$
$$Q_f = \left[f_\theta(\tilde{q}_1) + f_{pe}(1), \ldots, f_\theta(\tilde{q}_F) + f_{pe}(F)\right] \in \mathbb{R}^{F \times D}. \tag{16}$$

$f_\theta(\cdot) : \mathbb{R}^{C \times H \times W} \mapsto \mathbb{R}^D$ is the backbone for embedding each frame into a $D$-dimensional vector. While $f_{pe}(\cdot)$ represents the position embedding function, which can be a cosine/sine function or a learnable function, it is important to note that the function type can impact the performance of the model.

Three linear layers including $\Psi(\cdot), \Gamma(\cdot) : \mathbb{R}^{F \times D} \mapsto \mathbb{R}^{F \times d_k}$ and $\Lambda(\cdot) : \mathbb{R}^{F \times D} \mapsto \mathbb{R}^{F \times d_v}$ are applied for constructing the prototype of support $P^c$, refers to:

$$A^{ck} = \text{LN}\left(\Psi\left(Q_f\right)\right) \cdot \text{LN}\left(\Gamma\left(S_f^{ck}\right)\right),$$
$$P^c = \frac{1}{K} \sum_{k=1}^{K}\left(\text{Softmax}\left(A^{ck}\right) \cdot \Lambda\left(S_f^{ck}\right)\right). \tag{17}$$

The $\text{LN}(\cdot)$ means the layer normalization and $\text{Softmax}(\cdot)$ represents the Softmax function.

## 3.7 Training Objective

The linear layer $\Lambda(\cdot)$ is applied on $Q_f$ to ensure the same operation as $S_f^{ck}$. By obtaining the closest distance between $Q_f$ and $P^c$, model can predict the label $\tilde{y}$ of $Q$, as:

$$\tilde{y} = \arg\min_c \underbrace{\left\|P^c - \left(\Lambda\left(Q_f\right)\right)\right\|}_{D(Q, S^c)}. \tag{18}$$

The cross-entropy loss $\mathcal{L}_{ce}$ is selected as the objective loss for training, which can be calculated with the ground truth $y$:

$$\mathcal{L}_{ce} = -\frac{1}{N} \sum_{i=1}^{N} y_i \log(\tilde{y}_i). \tag{19}$$

## 4 Experiments

### 4.1 Experimental Configuration

*4.1.1 **Datasets Processing.*** We select four widely used datasets, *i.e.*, SthSthV2 [9], Kinetics [2], UCF101 [23], and HMDB51 [12]. Apart from SthSthV2, which is temporal-related, other three datasets are all spatial-related. When decoding videos, we set the sampling intervals to every 1 frame. For simulating various fluency, the intervals can be adjusted. Building upon data split from prior works [1, 38, 43], we split our datasets into training, validation, and testing sets. A task within each set, classifying query samples into one of the support classes serves as a training unit.

Following TSN [27], 8 frames ($F = 8$) resized to $3 \times 256 \times 256$ are uniformly and sparsely sampled each time. Data augmentations in training include random crops to $3 \times 224 \times 224$ and horizontal flipping, while only a center crop is used during testing. As pointed out in previous works [1], many action classes in SthSthV2 are sensitive to horizontal direction (*e.g.*, "Pulling S from left to right"[1]). Hence, we avoid horizontal flipping on this dataset.

---
[1]"S" refers to "something"

**Table 1: Performance (↑ Acc. %) comparison. "★": our implementation, "†": multimodal methods, "N/A": not available in the publication, Bold texts: the best results, underline texts: the previous best results.**

| Methods | Backbone | SthSthV2 | | Kinetics | | UCF101 | | HMDB51 | |
|---|---|---|---|---|---|---|---|---|---|
| | | 1-shot | 5-shot | 1-shot | 5-shot | 1-shot | 5-shot | 1-shot | 5-shot |
| CMN [43, 44] | ResNet-50 | 36.2 | 42.8 | 60.5 | 78.9 | N/A | N/A | N/A | N/A |
| OTAM [1] | ResNet-50 | 42.8 | 52.3 | 73.0 | 85.8 | N/A | N/A | N/A | N/A |
| ITANet [40] | ResNet-50 | 49.2 | 62.6 | 73.6 | 84.3 | N/A | N/A | N/A | N/A |
| TA$^2$N [13] | ResNet-50 | 47.6 | 61.0 | 72.8 | 85.8 | 81.9 | 95.1 | 59.7 | 73.9 |
| LSTC [16] | ResNet-50 | 47.6 | 66.7 | 73.4 | 86.5 | 85.7 | 96.5 | 60.9 | 76.8 |
| STRM [24] | ResNet-50 | N/A | 68.1 | N/A | 86.7 | N/A | 96.9 | N/A | 76.3 |
| HCL [42] | ResNet-50 | 47.3 | 64.9 | 73.7 | 85.8 | 82.6 | 94.5 | 59.1 | 76.3 |
| SloshNet [34] | ResNet-50 | 46.5 | 68.3 | N/A | 87.0 | N/A | 97.1 | N/A | 77.5 |
| SA-CT [41] | ResNet-50 | 48.9 | 69.1 | 71.9 | 87.1 | 85.4 | 96.4 | 60.4 | 78.3 |
| GCSM [37] | ResNet-50 | N/A | N/A | 74.2 | 88.2 | 86.5 | 97.1 | 61.3 | 79.3 |
| GgHM [35] | ResNet-50 | 54.5 | 69.2 | 74.9 | 87.4 | 85.2 | 96.3 | 61.2 | 76.9 |
| ★TRX [18] | ResNet-50 | 55.8 | 69.8 | 74.9 | 85.9 | 85.7 | 96.3 | 66.5 | 77.2 |
| ★HyRSM [30] | ResNet-50 | 54.1 | 68.7 | 73.5 | 86.2 | 83.6 | 94.6 | 60.1 | 76.2 |
| ★MoLo [29] | ResNet-50 | 56.6 | 70.7 | 75.2 | 85.7 | 86.2 | 95.4 | 67.1 | 77.3 |
| †AmeFu-Net [8] | ResNet-50 | N/A | N/A | 74.1 | 86.8 | 85.1 | 95.5 | 60.2 | 75.5 |
| †MTFAN [33] | ResNet-50 | 45.7 | 60.4 | 74.6 | 87.4 | 84.8 | 95.1 | 59.0 | 74.6 |
| †AMFAR [31] | ResNet-50 | 61.7 | 79.5 | 80.1 | 92.6 | 91.2 | 99.0 | 73.9 | 87.8 |
| ★†Lite-MKD [14] | ResNet-50 | 55.7 | 69.9 | 75.0 | 87.5 | 85.3 | 96.8 | 66.9 | 74.7 |
| SOAP-Net (Ours) | ResNet-50 | **61.9** | **79.8** | **81.1** | **93.8** | **94.1** | **99.3** | **76.5** | **88.4** |
| STRM [24] | ViT-B | N/A | 70.2 | N/A | 91.2 | N/A | 98.1 | N/A | 81.3 |
| SA-CT [41] | ViT-B | N/A | 66.3 | N/A | 91.2 | N/A | 98.0 | N/A | 81.6 |
| ★TRX [18] | ViT-B | 57.2 | 71.4 | 76.3 | 87.5 | 88.9 | 97.2 | 66.9 | 78.8 |
| ★HyRSM [30] | ViT-B | 58.8 | 71.3 | 76.8 | 92.3 | 86.6 | 96.4 | 69.6 | 82.2 |
| ★MoLo [29] | ViT-B | 61.1 | 71.7 | 78.9 | 95.8 | 88.4 | 97.6 | 71.3 | 84.4 |
| †CLIP-FSAR [28] | ViT-B | 61.9 | 72.1 | 89.7 | 95.0 | 96.6 | 99.0 | 75.8 | 87.7 |
| SOAP-Net (Ours) | ViT-B | **66.7** | **81.2** | **89.9** | **95.5** | **96.8** | **99.5** | **79.3** | **89.8** |

### 4.1.2 *Implementation Details and Evaluation.*
In 3DEM, the size of Conv3D is $3 \times 3 \times 3$. For CWEM, the expand channel number $C_r$ is set to 16, while Conv1D size is 3. We set $O = \{1, 2, 3\}$ in HMEM, the Conv2D$^M$ with shared parameters are also $3 \times 3$ in each branch. All 2D convolution for changing channel number is set to $1 \times 1$. Except for these channel recovery convolutions, other convolutions maintain the same shape of input and output. We employ widely-used ResNet-50 [10] or ViT-B [6] as the backbone initialized with pre-trained weights on ImageNet [5]. The final outputs of the backbone are 2048-dimensional vectors ($D = 2048$). For three linear layers, the parameters are randomly initialized while $d_k$ and $d_v$ are both set to 1152.

We utilize the standard 5-way 5-shot and 1-shot few-shot settings. Due to its larger size, SthSthV2 requires 75,000 tasks for training, while other datasets employ 10,000 tasks. An SGD optimizer is applied to train our model, with an initial learning rate of $10^{-3}$. The training process is conducted on a deep learning server equipped with two NVIDIA 24GB RTX3090 GPUs. Validation set determines hyperparameters. During testing, we report average accuracy across 10,000 random tasks from the testing set.

## 4.2 Comparison with Various Methods

### 4.2.1 *Comparison with ResNet-50 Backbone Methods.*
Based on the average accuracy reported in Table 1, we have the following observation. Our SOAP-Net outperforms other methods and achieves SOTA performance. For example, on the Kinetics dataset

under the 1-shot setting, SOAP improves the current SOTA performance of MoLo [29] from 75.2% to 81.1%. We believe the motion information within adjacent frames is insufficient, despite MoLo introducing it. Similar improvements are also found in other datasets under diverse few-shot settings, showing that spatio-temporal relation and comprehensive motion information from SOAP lead to significant improvement in FSAR. It is worth mentioning that SOAP-Net even surpasses multimodal methods.

### 4.2.2 *Comparison with ViT-B Backbone Methods.*
ResNet-50 serves as the backbone network in most previous works, while ViT-B is rarely used in FSAR. For a comprehensive comparison, we implemented several methods and replaced their backbones with ViT-B, finding it outperformed ResNet-50 in FSAR due to larger model capacity. The previous SOTA performance is also achieved by MoLo, benefiting from optimizing spatio-temporal relation construction and comprehensive motion information. A similar trend is observed on ResNet-50 and ViT-B. These comparisons reveal the competitiveness of our proposed method.

## 4.3 Essential Components and Factors

### 4.3.1 *Analysis of Key Components.*
We first divide SOAP into three key components: 3DEM, CWEM, and HMEM. Experiments on individual or combined components are conducted on two datasets under diverse few-shot settings. As shown in Table 2, results indicate that each SOAP component can improve FSAR performance. However, the most significant boost comes from HMEM, likely due

**Table 2: Analysis (↑ Acc. %) of Key Components.**

| 3DEM | CWEM | HMEM | SthSthV2 | | Kinetics | |
|---|---|---|---|---|---|---|
| | | | 1-shot | 5-shot | 1-shot | 5-shot |
| ✗ | ✗ | ✗ | 54.5 | 67.3 | 74.1 | 85.2 |
| ✔ | ✗ | ✗ | 55.6 | 69.4 | 76.8 | 86.6 |
| ✗ | ✔ | ✗ | 55.4 | 70.2 | 76.1 | 86.1 |
| ✗ | ✗ | ✔ | **58.3** | **72.3** | **78.5** | **88.9** |
| ✔ | ✔ | ✗ | 58.5 | 73.1 | 78.8 | 89.3 |
| ✗ | ✔ | ✔ | 60.5 | 77.8 | 79.0 | 92.2 |
| ✔ | ✗ | ✔ | **60.7** | **78.5** | **80.2** | **92.6** |
| ✔ | ✔ | ✔ | **61.9** | **79.8** | **81.1** | **93.8** |

to fuller motion information usage. Combining two components with HMEM performs better than without HMEM. We conclude that comprehensive motion information plays a more significant role in FSAR. Using all SOAP components yields the best results, proving all are essential.

*4.3.2  **Frame Tuples and Branches Design.*** Results under various hyperparameter $O$ settings of HMEM is demonstrated in Table 3. Each element represents frame counts, with the cardinality of set $|O|$ equaling number of branches. Performance generally improves with more branches. Specifically, any frame tuple size improves with HMEM having one branch. Better results occur with two branches. Best performance occurs at 3 branches under $O = \{1, 2, 3\}$ setting. This aligns with intuition comprehensive motion information contributes more for FSAR. However, excessive branches risk degradation. Too much motion information overlap does not provide useful information. For example, $O = \{4\}$ acts similarly to two $O = \{2\}$ configurations together, implying that they perform worse than other non-overlapping settings. While $O = \{1\}$ provides motion information between frames not conflicting with other $O$ settings for building frame tuples, introducing comprehensive motion information.

**Table 3: Impact (↑ Acc. %) of $O$ Design.**

| $O$ Design | SthSthV2 | | Kinetics | |
|---|---|---|---|---|
| | 1-shot | 5-shot | 1-shot | 5-shot |
| $O = \{1\}$ | 59.1 | 73.9 | 78.9 | 89.5 |
| $O = \{2\}$ | **59.5** | 74.3 | 79.1 | **90.8** |
| $O = \{3\}$ | 59.4 | **74.6** | **79.5** | **90.8** |
| $O = \{4\}$ | **59.5** | 74.2 | 79.3 | 90.5 |
| $O = \{1, 2\}$ | 60.6 | 77.6 | 80.3 | 92.3 |
| $O = \{1, 3\}$ | **60.7** | 77.8 | 80.1 | 92.4 |
| $O = \{1, 4\}$ | 60.4 | 78.2 | **80.5** | 92.4 |
| $O = \{2, 3\}$ | **60.7** | **78.3** | **80.5** | 92.1 |
| $O = \{2, 4\}$ | 60.1 | 77.2 | 80.0 | 91.9 |
| $O = \{3, 4\}$ | 60.4 | **78.3** | 80.2 | **92.9** |
| $O = \{1, 2, 3\}$ | **61.9** | **79.8** | **81.1** | **93.8** |
| $O = \{1, 2, 4\}$ | 60.7 | 77.9 | 80.5 | 92.6 |
| $O = \{1, 3, 4\}$ | **61.9** | 79.3 | 80.7 | 93.5 |
| $O = \{2, 3, 4\}$ | 60.7 | 77.7 | 80.3 | 92.3 |
| $O = \{1, 2, 3, 4\}$ | 61.5 | 79.2 | 80.7 | 92.8 |

*4.3.3  **The Impact of Temporal Order.*** Up to this point, we believe the frame tuple should follow the temporal order for better

**Table 4: Impact (↑ Acc. %) of Temporal Order.**

| Reversed Order | SthSthV2 | | Kinetics | |
|---|---|---|---|---|
| | 1-shot | 5-shot | 1-shot | 5-shot |
| ✔ | 54.1 | 66.6 | 79.2 | 91.6 |
| ✗ | **61.9** | **79.8** | **81.1** | **93.8** |

representation. After determining $O = \{1, 2, 3\}$ setting, we conduct experiments with temporal and reversed order. In an extreme scenario, the frames in support take the temporal order while frames in query are reversely ordered during inference. As the results in Table 4, SthSthV2 and Kinetics datasets both have a drop with reversed order settings. However, the drop in SthSthV2 is much larger than in Kinetics. The dataset type supports our previous observation that the temporal-related SthSthV2 is more sensitive to order than the spatial-related Kinetics. Based on above results, we can infer that SOAP is closely tied to temporal order.

## 4.4  Research on How SOAP Works

*4.4.1  **CAM Visualization.*** The CAM [21] for "crossing river" from the Kinetics dataset is shown in Figure 6. The first row shows raw HFR video frames. We observe that the timeline and displacement of the moving peoples are very subtle, making it difficult to detect the spatio-temporal relation and motion information. In the second row, we see that the model attends more to the background than moving peoples. This result supports our observation in HFR videos. With the help of SOAP, we detect that the focus shifts from the background to the moving peoples, despite the subtle timeline and displacement. Based on this analysis of the CAM, we conclude that motion information is crucial in FSAR. Spatio-temporal relation and comprehensive motion information from SOAP can significantly improve feature representation.

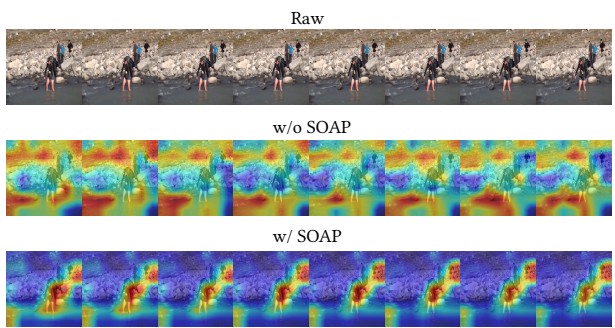

Raw

w/o SOAP

w/ SOAP

**Figure 6: Example of the "crossing river" selected from Kinetics, timeline is from left to right.**

*4.4.2  **T-SNE Visualization.*** In the 5-way task, feature distributions can be visualized using t-SNE [25]. We select five representative yet challenging support action classes from Kinetics: "ski jumping", "snowboarding", "skateboarding", "crossing river", and "driving tractor". The t-SNE visualization is shown in Figure 7, with different colors and markers representing distinct action classes. Without SOAP (Figure 7(a)), the five classes are generally separable but with some overlapping, such as in the right ("driving tractor"

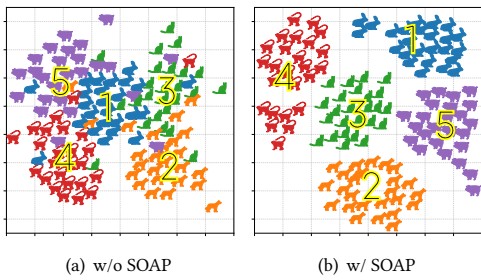

Figure 7: T-SNE visualization of five action classes in support from Kinetics. **Blue Rabbits**: "ski jumping", **Orange Dogs**: "crossing river", **Green Cats**: "driving tractor", **Red Monkeys**: "snowboarding", **Purple Pigs**: "skateboarding".

and "crossing river"), left ("skateboarding" and "snowboarding"), and center ("ski jumping"). By contrast, with SOAP (Figure 7(b)), the five classes are more distinctly separated and same-class samples are more tightly clustered. This strong evidence demonstrates the enhanced representational ability of SOAP for support features.

## 4.5 Pluggability of SOAP

*4.5.1 On RGB-Based Methods.* In Table 5, we experimentally demonstrate that the SOAP generalizes well to other methods by inserting it into widely used methods including TRX, HyRSM, and MoLo. Using Kinetics as an example, TRX benefits from comprehensive motion information and achieves 7.6% gains in the 1-shot setting and 8.5% in the 5-shot setting. Similar improvements are also observed in the other two methods. These results fully prove that optimizing spatio-temporal relation and comprehensive motion information are particularly useful for feature extraction.

Table 5: Pluggability (↑ Acc. %) on RGB-Based Methods.

| RGB-Based Methods | SthSthV2 | | Kinetics | |
|---|---|---|---|---|
| | 1-shot | 5-shot | 1-shot | 5-shot |
| TRX | 55.8 | 69.8 | 74.9 | 85.9 |
| SOAP-TRX | **62.3** | **80.3** | **82.5** | **94.4** |
| HyRSM | 54.1 | 68.7 | 73.5 | 86.2 |
| SOAP-HyRSM | **60.1** | **75.8** | **79.9** | **91.6** |
| MoLo | 56.6 | 70.7 | 75.2 | 85.7 |
| SOAP-MoLo | **60.9** | **76.6** | **79.6** | **92.1** |

*4.5.2 On Multimodal Methods.* Backbone networks serve as feature extractors in multimodal methods for FSAR. Additional information from other modalities enhances performance. In most multimodal methods, depth or optical flow information augments RGB data. Apart from the RGB backbone, additional depth-specific or optical-flow-specific networks are also utilized. As depth and optical flow are both derived from raw RGB frames, they inherently contain spatio-temporal relation and motion information. Therefore, we hypothesize SOAP can also benefit multimodal methods and implement several multimodal methods equipped with SOAP for evaluation. Results in Table 6 demonstrate that all selected multimodal methods achieve further improvement via optimized

Table 6: Pluggability (↑ Acc. %) on Multimodal Methods.

| Multimodal Methods | SthSthV2 | | Kinetics | |
|---|---|---|---|---|
| | 1-shot | 5-shot | 1-shot | 5-shot |
| AmeFu-Net | 56.2 | 71.1 | 76.3 | 88.2 |
| SOAP-AmeFu-Net | **62.5** | **80.4** | **82.2** | **94.6** |
| MTFAN | 55.8 | 70.6 | 74.8 | 87.8 |
| SOAP-MTFAN | **62.2** | **81.3** | **81.6** | **93.7** |
| Lite-MKD | 55.7 | 69.9 | 75.0 | 87.5 |
| SOAP-Lite-MKD | **61.8** | **80.6** | **81.1** | **93.2** |
| AMFAR | 61.3 | 78.9 | 79.3 | 91.4 |
| SOAP-AMFAR | **64.7** | **82.6** | **83.9** | **95.3** |

spatio-temporal relation construction and comprehensive motion information capturing from SOAP. These results provide legitimate validation for our hypothesis and emphasize the advanced pluggability of SOAP on multimodal methods.

## 4.6 Generalization Study

In this study, we aim to simulate different levels of frame-rate by varying the sampling interval from a minimum of 1 to a maximum of 6 and conduct a series of experiments under a well-established 5-way 5-shot setting. As shown in Figure 8, the performance tends to decrease gradually with the increase of frame-rate. We find that recent FSAR methods all experience drastic degradation in performance with HFR videos while SOAP-Net indicates performance stability, demonstrating the superiority of SOAP across varying frame-rate. These phenomena also highlight the key role of spatio-temporal relation optimization and comprehensive motion information in advancing FSAR performance.

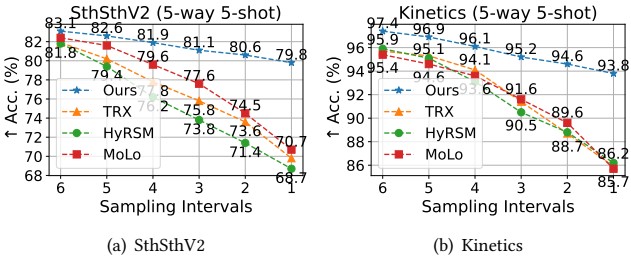

Figure 8: Performance (↑ Acc. %) of Various Frame-Rates.

## 5 Conclusion

In this paper, we propose the plug-and-play SOAP to optimize spatio-temporal relation construction and capture comprehensive motion information for FSAR. SOAP takes into account temporal connections across feature channels and spatio-temporal relation within features. It combines frame tuples of various frame counts to provide a broader perspective for motion information. Considering the competitiveness, pluggability, generalization, and robustness of SOAP, we hope and believe that our work will offer valuable insights for future research in multimedia analytic.

## Acknowledgments

The authors would like to appreciate all participants of peer review. Wenbo Huang sincerely thanks Bingxiao Shi (USTB) and all family members for the encouragement at an extremely difficult time. This work is supported by the National Key Research and Development Program for the 14th-Five-Year Plan of China 2023YFC3804104 in 2023YFC3804100, National Natural Science Foundation of China under Grants No.62072099, 62232004, 62373194, 62276063, Jiangsu Provincial Key Laboratory of Network and Information Security under Grant BM2003201, Key Laboratory of Computer Network and Information Integration of MOE China under Grant No.93K-9, and the Fundamental Research Funds for the Central Universities.

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
