# OpenReview forum: "SOAP: Enhancing Spatio-Temporal Relation and Motion Information Capturing for Few-Shot Action Recognition"
_acmmm.org/ACMMM/2024/Conference — MM2024 Poster_

### Official Review · Reviewer_hutr · 2024-05-18

**Rating:** 4
**Confidence:** 3

**Summary:**

The paper introduces a novel plug-and-play architecture called SOAP-Net, designed to enhance spatio-temporal relation and motion information capturing for few-shot action recognition (FSAR). The authors identify two main challenges in FSAR: optimizing spatio-temporal relation construction and comprehensive motion information capturing. To address these, SOAP-Net employs a 3D convolution-based module for spatio-temporal relation construction, a channel-wise enhancement module for temporal connections, and a hybrid motion enhancement module for capturing motion information across multiple frames. The model achieves state-of-the-art performance on benchmarks such as SthSthV2, Kinetics, UCF101, and HMDB51.

**Strengths:**

## Strengths

Novelty: SOAP-Net presents a novel approach to FSAR by integrating spatial and temporal features in a more cohesive manner than previous methods. The use of frame tuples and a hybrid approach for motion information is particularly innovative.

Technical Approach:  The paper's technical approach is sound, with a well-defined problem formulation and a clear explanation of the SOAP-Net architecture. The use of 3D convolutions and channel-wise enhancements is theoretically justified and aligns with the current trends in action recognition research.

Evaluation: The authors provide an extensive experimental evaluation, demonstrating SOAP-Net's effectiveness across multiple datasets. The comparison with state-of-the-art methods and ablation studies further validates the model's performance.

Clarity: The paper is well-organized, with a clear presentation of the methodology, experiments, and results. The figures and tables are informative and enhance the reader's understanding of the proposed approach.

Applications: The potential applications of SOAP-Net are broad, including intelligent surveillance, video understanding, and health monitoring, which demonstrates the practical relevance of the research.

**Limitations:**

CWEM is essentially a form of channel attention mechanism, and there is a lack of experimental comparison with other channel or spatial attention mechanisms such as SENet and CBAM.

SENet: Hu J, Shen L, Sun G. Squeeze-and-excitation networks[C]//Proceedings of the IEEE conference on computer vision and pattern recognition. 2018: 7132-7141.
CBAM： Woo S, Park J, Lee J Y, et al. Cbam: Convolutional block attention module[C]//Proceedings of the European conference on computer vision (ECCV). 2018: 3-19.

**Suitability:**

3

---

### Official Review · Reviewer_pL9f · 2024-05-20

**Rating:** 3
**Confidence:** 4

**Summary:**

In order to better construct spatiotemporal relations and focus on motion information in FSAR, the paper designs spatial, channel, and temporal attention models. The proposed method achieves new SOTA performance on several well-known FSAR benchmarks.

**Strengths:**

-  Technical correctness.
- Evaluation: the proposed method achieves new SOTA performance on several well-known FSAR benchmarks. Comprehensive experiments demonstrate the competitiveness, pluggability, generalization, and robustness of the proposed method.
- Clear presentation.

**Limitations:**

-  Novelty: 1. This article places attentions before the backbone, which is different from the previous methods. Is this a reason for the superior performance of the proposed method? In other words, just adding other attention before the backbone will produce good results? 2. The authors' use of prototype construction is also different from the previous methods . And in Table 2, only using prototype construction surpasses many methods. Does this mean that the outstanding performance of proposed method comes from the prototype construction?

**Suitability:**

2

---

### Official Review · Reviewer_h7sF · 2024-05-24

**Rating:** 5
**Confidence:** 3

**Summary:**

This paper proposes a novel plug-and-play architecture called Spatio-temporal frame tuple enhancer (SOAP) for few-shot action recognition. In this paper, temporal connections between different feature channels and spatio-temporal relation of features are considered instead of simple feature extraction.

**Strengths:**

SOAP-Net achieves new state-of-the-art performance across well-known benchmarks such as SthSthV2, Kinetics, UCF101, and HMDB51. Extensive empirical evaluations underscore the competitiveness, pluggability, generalization, and robustness of SOAP.

**Limitations:**

1. The timeline and displacement of object in HFR video frames are much subtler than those in low frame-rate (LFR) video frames, better reflecting fine-grained actions. However, the spatio-temporal relation and motion information density decrease with the improvement of video fluency. How can this method extract accurately fine-grained features from high motion information density LFR videos with sparse fine-grained features?
2. What multimodal tasks does the multimodal methods address in Section 4.5.2?
3. In Figure 6, the difference between the video frames is very small. In this case, how can the model capture the temporal features or relationships of the videos?

**Suitability:**

3

---

### Meta-Review · Area_Chair_1FCM · 2024-06-30

**Recommendation:** Accept (Poster)
**Confidence:** 5

**Metareview:**

This work is focused on few-shot action recognition. It received weak accept, borderline reject and borderline accept ratings initially and the authors provided a rebuttal. After the rebuttal most of the concerns were addressed and the reviewers provide 2x weak accept and borderline accept as final ratings. The AC agree with the assessment and recommend this work for acceptance. The authors should address the comments from the reviewers in preparing the final submission.